# Nonlinear Piezoresistive Behavior of Plain-Woven Carbon Fiber Reinforced Polymer Composite Subjected to Tensile Loading

**Ignatius Pulung Nurprasetio [1,\*], Bentang Arief Budiman [1,2], Ahmad Alfin Afwan [1], Putri Nur Halimah [1,2], Sarah Tania Utami [1] and Muhammad Aziz [3,†]**

[1] Faculty of Mechanical and Aerospace Engineering, Institut Teknologi Bandung, Ganesha no. 10, Bandung 40132, Indonesia; bentang@ftmd.itb.ac.id (B.A.B.); alfin.afwan@gmail.com (A.A.A.); putri_nurhalimah@yahoo.co.id (P.N.H.); sarahtaniautami@yahoo.com (S.T.U.)

[2] National Center for Sustainable Transportation Technology, Ganesha no. 10, Bandung 40132, Indonesia

[3] Institute of Innovative Research, Tokyo Institute of Technology, 2-12-1 Ookayama, Meguro-ku, Tokyo 152-8550, Japan; maziz@iis.u-tokyo.ac.jp

\* Correspondence: ipn@ftmd.itb.ac.id; Tel.: +62-22-2504243

† Current address: Institute of Industrial Science, The University of Tokyo, 4-6-1 Komaba, Meguro-ku, Tokyo 153-8505, Japan.



**Featured Application: Authors are encouraged to provide a concise description of the specific application or a potential application of the work. This section is not mandatory.**

**Abstract:** This work aims to investigate piezoresistive behavior in plain-woven carbon fiber reinforced polymer (CFRP). Measurement method for electric resistant alteration in the woven CFRP under tensile loading by using a Wheatstone bridge circuit is introduced. Reversibility of the resistant alteration is also investigated whereas the gauge factor of the woven CFRP is evaluated. The result shows that the positive piezoresistive properties of the woven CFRP can be observed by the Wheatstone bridge circuit. The specific resistances of 43.8 μΩm and 10.1 μΩm are obtained for wrap and thickness directions, respectively. Reversibility with a hysteresis of the woven CFRP can also be confirmed with the gauge factor of 22.9 at loading conditions and 17.7 at unloading conditions. Positive piezoresistive behavior which has been revealed in this work can be utilized for structural health monitoring technology development.

**Keywords:** plain-woven CFRP; piezoresistivity; wheatstone bridge; structural health monitoring

## 1. Introduction

Woven carbon fiber reinforced polymer (CFRP) has become attractive to be used for critical load-bearing structures in airplanes, trains, ships, and cars. The woven pattern can improve structural stability since the loads on wrap and weft directions can be held by fibers simultaneously in one lamina [1]. As a consequence, it can increase the damage tolerance of the CFRP structure. Compared to unidirectional CFRP, the manufacturing process of woven CFRP are relatively easy and fast [2].

One of the main challenges to use woven CFRP is the difficulty in detecting structural damages due to intricate patterns and anisotropic properties [3]. The damages need to be detected as early as possible to avoid structural failures during operation. The development of structural health monitoring (SHM) technology for woven CFRP that can detect initial damages needs to be conducted. SHM aims to create a time domain-based structural failure monitoring system that can be used as the basis for establishing a diagnosis or prognosis [4]. SHM has been developed for unidirectional CFRP in the last

few decades. The most popular SHM technology uses the piezoresistive behavior of the CFRP to detect the damages [5]. The behavior enables CFRP to be like a strain gauge in which its electrical resistance changes when mechanical strain increases due to exposure of the load. By measuring the electrical resistance alteration, the strain condition in the CFRP can be investigated. Furthermore, the level and type of damages to the CFRP may be determined by analyzing the recorded electrical resistance [6,7].

Some researchers report that piezoresistive behavior can be measured by a four-wire method [8,9]. Schulte and Baron stated that unidirectional CFRP has positive piezoresistivity in which electrical resistance increases proportionally with the increased mechanical strain [10]. This result is also supported by the findings reported by Abry et al. [11] and Angelidis et al. [12]. However, different results were obtained by Wang et al. [9,13] and Mei et al. [14]. By using two probe-wires and four probe-wires methods, they reported that the CFRP had a negative piezoresistivity. Both argued that the positive piezoresistivity that was reported by many researches was previously caused by increased electrical resistance in electrodes, not from the changes in electric resistance in CFRP.

Todoroki and Yoshida conducted tests with techniques similar to Wang et al. [9,13] and Mei et al. [14] and produced negative piezoresistivity [8]. However, they argued that the negative piezoresistivity occurred due to poor surface treatment prior to the electrode attachment. For fine surface treatment, the CFRP has positive piezoresistivity with gauge factor in a range between 2 to 4. The latest study was conducted by Jeon et al. to find out the effect of contact resistance between CFRP and electrodes to the piezoresistive behavior [15]. They also reported that the roughness of the composite surface attached to the electrodes affects the test results. Thus, these factors need to be considered in testing and application. Furthermore, the use of piezoresistivity can be extended to the development of self-healing technologies, namely the ability of the CFRP to repair the damage to their structures. The concept of self-healing using piezoresistivity has been revealed by Park et al. by utilizing the Diels-Alder reaction [16,17].

Unlike like piezoresistive behavior in the unidirectional CFRP that has been investigated intensively, the behavior in woven CFRP still needs to be further clarified. Indeed, the behavior is strongly influenced by various factors and parameters, including the fiber arrangement. The woven pattern might cause nonlinear piezoresistive behavior in line with its mechanical behavior [18,19]. The relationship between these parameters needs to be revealed for the future implementation and development of SHM technology.

In this work, the piezoresistive behavior in plain-woven CFRP is investigated. A method to investigate the relationship between electrical and mechanical strains is proposed by integrating Wheatstone bridge circuits, strain gauge, and electrodes. This method is different as compared to the common two or four-wire methods. Through this method, the piezoresistive behavior of the woven CFRP is clarified, including its nonlinearity. The values of the gauge factor and the reversibility of the piezoresistive behavior are also calculated by introducing an electrical circuit model of the CFRP. The application of the piezoresistive behavior in the CFRP for developing structural health monitoring is also comprehensively discussed.

## 2. Modeling of Piezoresistive Behavior

### 2.1. Nonlinear Piezoresistive Behavior on Plain-Woven CFRP

In CFRP, electrical current can flow via carbon fibers [20]. The arrangement of the fibers inside the CFRP determines the electrical resistance. This is also the reason why the woven CFRP has different electrical characteristics with unidirectional CFRP. Figure 1 shows a schematic illustration of fiber direction in plain-woven CFRP. The fibers initially have a certain waving position, which depends on manufacturing process quality. Under tensile loading, fibers in wrap direction change their arrangement from waving to straightening prior to tightening. In contrast, under tensile loading, fibers in the weft direction approach to fibers in wrap direction and spread along the wrap direction. The rearrangement of these fibers may cause the electrical resistance changes differently compared to

unidirectional CFRP. Fiber distribution in the weft direction plays an important role in determining overall electrical resistance.

During the unloading process, both fibers in wrap and weft directions may not return back to the initial waving position, especially after imposing high tensile loading, which causes high strain conditions. Depending on the rearrangement of those fibers, nonlinear behavior and residual electric resistance might appear during the unloading process. This behavior must be evaluated accurately to improve the design process of the SHM for the woven CFRP.

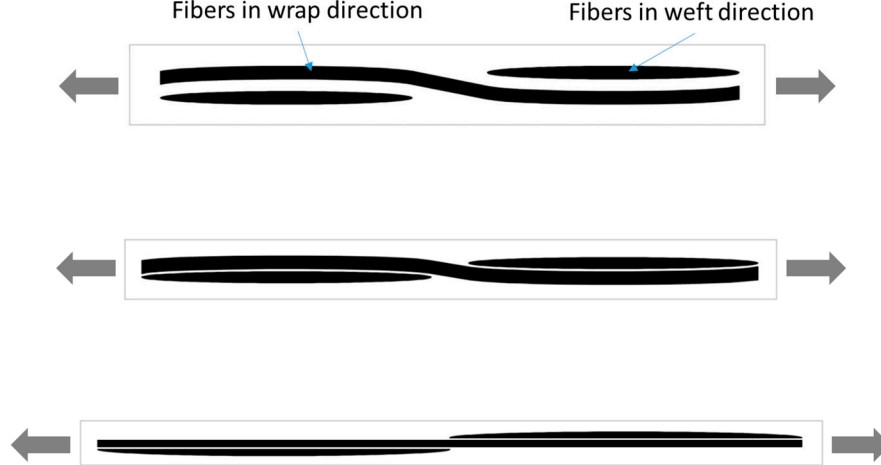

**Figure 1.** Illustration of the fiber arrangement in plain-woven carbon fiber reinforced polymer (CFRP) under tensile test.

*2.2. Electrical Circuit Model for Plain-Woven CFRP*

SHM on the woven CFRP is usually implemented by installing electrodes forming a network on the top and bottom surfaces. The electrodes flow electric current to monitor the electric resistance of the CFRP, which directly corresponds to deformation or damages. In this work, plain-woven CFRP is used as a specimen. This CFRP has the simplest fiber pattern compared to other woven composite types in which it has symmetric properties in wrap and weft directions. Thus, the characteristic of plain-woven CFRP can be modeled as an electrical circuit with electrical resistances along with the wrap and thickness directions.

The model that represents the electrical resistances in the CFRP is shown in Figure 2a. Two electrodes are installed on the top surface of the plain-woven CFRP with a certain distance ($L_c$) to investigate electric resistance alteration due to deformation. There is contact resistance ($R_c$) at the interface between these electrodes and composite surfaces. The $R_c$ depends on surface treatment prior to the attachment of electrodes. Since the plain-woven CFRP has identic geometry for each lamina, the resistance of each lamina for thickness direction ($R_t$) has identic value. Furthermore, considering each lamina in plain-woven CFRP has identic geometry in wrap and weft directions, the electric resistance along wrap direction ($R_w$) can also be considered having similar values with resistance along weft direction.

Total resistance ($R_{Total}$) among two electrodes installed on the top surface of the woven CFRP can be calculated as follows:

$$R_{Total} = R_{CFRP} + 2R_c, \tag{1}$$

where $R_{CFRP}$ is the total resistance of the CFRP. It can be calculated as follows:

$$R_{CFRP} = \frac{\left(2R_{t(n-1)} + R_{(n-1)}\right)R_{t(n)}}{R_{w(n)} + 2R_{t(n-1)} + R_{(n-1)}}, \tag{2}$$

where $n$ denotes the number of CFRP lamina.

In case that the electrodes are attached on the top and bottom surfaces of the woven composite, a series $R_t$ model can be implemented as shown in Figure 2b. In this model, the thickness of the specimen $(t_{(n)})$ is the addition of lamina thicknesses that have identic value. The $R_{CFRP}$ can be then calculated as follows:

$$R_{CFRP} = nR_t, \tag{3}$$

Electrical resistivity of lamina n in wrap direction $(\rho_{w(n)})$ and thickness direction $(\rho_{t(n)})$ can be then calculated as follows:

$$\rho_{w(n)} = \frac{R_{w(n)}t_{(n)}b}{L_c}, \tag{4}$$

$$\rho_{t(n)} = \frac{R_{t(n)}L_c b}{2t_{(n)}}, \tag{5}$$

where $b$ is width of specimen.

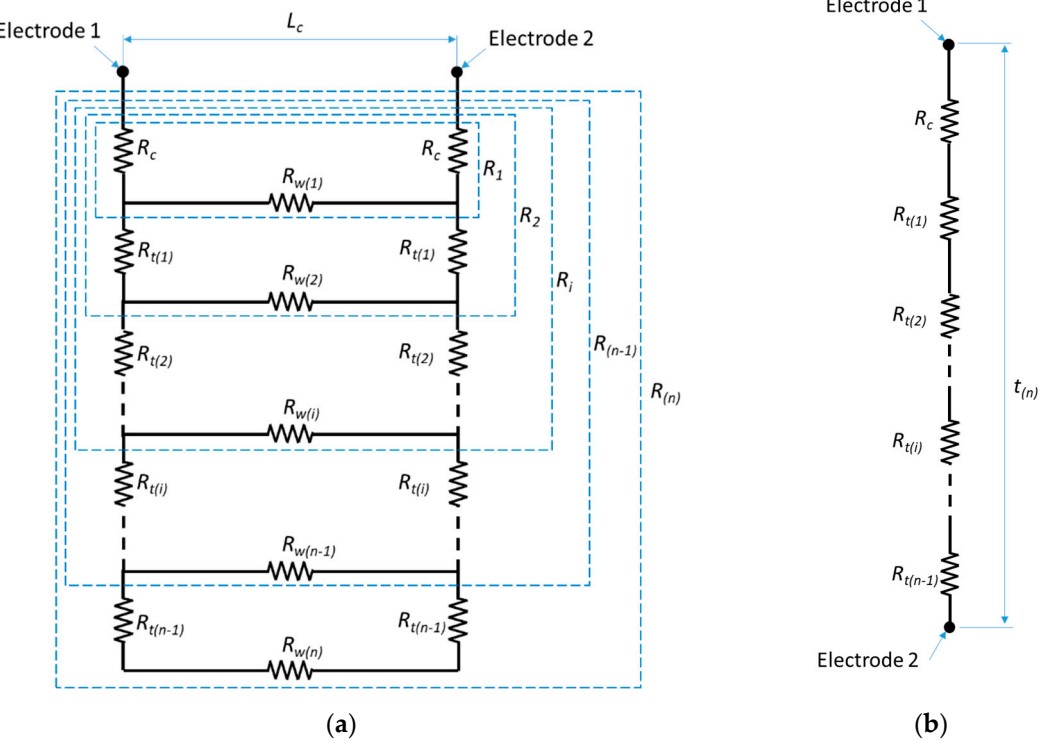

**Figure 2.** (**a**) Electrical circuit to laminate the plain-woven composite with both electrodes installed on the top surface and (**b**) electrodes installed on the top and bottom surface.

Another important parameter required for implementing SHM in the woven CFRP is the gauge factor $(G_{CFRP})$. It can be calculated as follows:

$$G_{CFRP} = \frac{\varepsilon_e}{\varepsilon_m}, \tag{6}$$

where $\varepsilon_e$ and $\varepsilon_m$ are electric and mechanical strains, respectively. Those can be calculated as follows:

$$\varepsilon_e = \frac{\Delta R}{(R_{CFRP} - \Delta R)}, \tag{7}$$

$$\varepsilon_m = \frac{\Delta L}{(L_c - \Delta L)}, \tag{8}$$

where $\Delta R$ is an electrical resistance alteration and $\Delta L$ is a deformation of the specimen. Positive value of $G_{CFRP}$ indicates positive piezoresistive behavior and vice versa. In case that nonlinear piezoresistive behavior appears, the $G_{CFRP}$ under loading and unloading conditions would have different values. The above difference must be investigated experimentally for developing SHM. The selection of $G_{CFRP}$ value applied to the electrical resistance model of the plain-woven composite must consider the nonlinearity and the existence of residual resistance.

## 3. Experimental Method

### 3.1. Specimen Preparation

A specimen of plain-woven CFRP produced by the pultrusion process was prepared for tensile testing. The specimen has six layers of the lamina with a total thickness of 2 mm. The used matrix composite was Epoxy resin. The CFRP was prepared by cutting the plate to follow the geometry of the tensile test specimen as depicted in Figure 3. Since the CFRP has orthotropic properties, the cutting line must be parallel to the fiber wrap direction. A strain gauge and electrodes were then installed on the specimen surface. Prior to the installation, the surface in which we connect the electrodes was first polished to minimize contact resistance between the electrodes and the specimen. The surface is polished gradually with sandpaper grades of P100, P120, and P180 for 3 min. The electrodes were then attached to the plain-woven CFRP surface by using silver paste and cyanoacrylate glue. Furthermore, the strain gauges were also attached to the backside of the specimen surface by using cyanoacrylate glue.

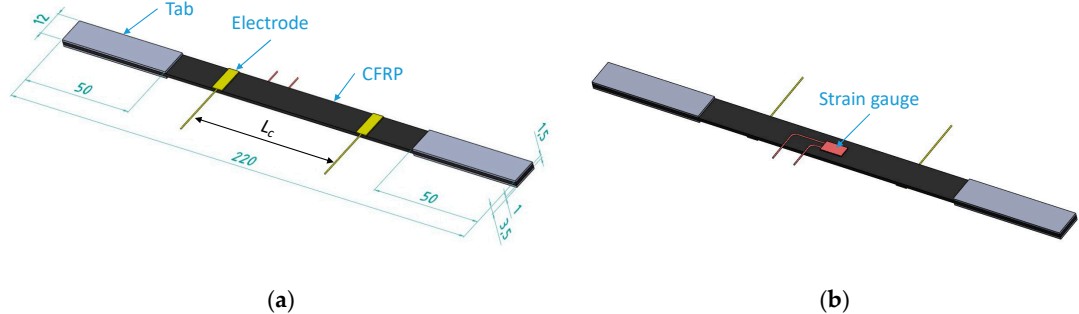

(**a**)                   (**b**)

**Figure 3.** The geometry of specimen with installed (**a**) electrodes and (**b**) strain gauge.

To investigate the effects of polishing to CFRP's tensile strength, specimens with dimensions shown in Figure 3a are made. The specimens were attached with a grip and pen made from stainless steel. These grips were attached to the sample using Epoxy resin to increase the strength of the specimens on the gripped area. Afterward, the specimens were polished using several types of sandpaper i.e., unpolished, P220, P600, and P2000.

The effects of polishing on the contact resistance of CFRP are also investigated. First, the surface of the spots that were going to be attached to an electrode was polished for minimizing the contact resistance that arises between electrodes and specimens. The surface was polished gradually with sandpaper grades of P0 (unpolished), P220, P600 and P2000. Electrodes are then attached to the plain-woven CFRP surface by using silver paste and cyanoacrylate glue.

### 3.2. Evaluating the Electrical Resistance of Plain-Woven CFRP

The $R_{CFRP}$ values in wrap and thickness directions were firstly evaluated by using the four-wire method without being imposed by tensile loading. These values were used for evaluating $\rho$ and further analysis of finding piezoresistive behavior. The schematics of electrical circuits of the four-wire method are shown in Figure 4a,b, respectively. The power supply has a role in supplying electric power to direct current (DC) signal generator GW Instek GPS-2303. The signal generator circulates the constant

DC ($I_{DC}$) to the electrodes attached to the specimen surface. The current passing through the resistance in this specimen was then captured as a DC voltage ($V_{DC}$) by the analog-to-digital converter (ADC) type PicoScope® 2204. The ADC serves to convert analog $V_{DC}$ signals into the digital signals so that they can be recorded by the data recorder. $R_{CFRP}$ can then be calculated as follows,

$$R_{CFRP} = \frac{V_{DC}}{I_{DC}} - 2R_c, \tag{9}$$

To assure a reliable measurement of the $R_{CFRP}$ values, 17 different constant $I_{DC}$ values with range from 4 to 20 mA are implemented.

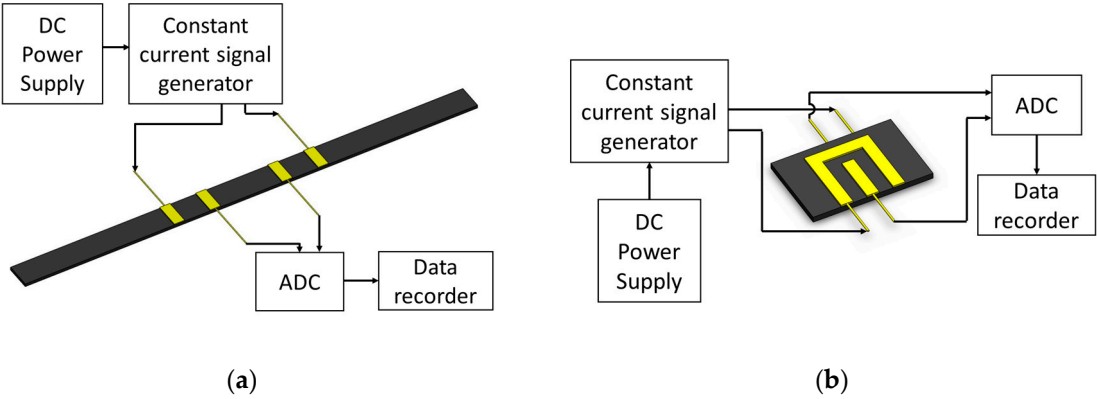

(**a**)                                              (**b**)

**Figure 4.** The electrical circuit for initial resistance evaluation in (**a**) wrap and (**b**) thickness directions.

### 3.3. Relationship between Electrical and Mechanical Strains

A series of tensile tests were conducted to reveal a relationship between electrical strain and mechanical strain in plain-woven CFRP. A tensile testing machine TENSILON RTF 1310 was used to apply quasi-static loading by stretching the specimen of plain-woven CFRP with a constant displacement speed of 2 mm/min. After the specimen was stretched to the strain of 350 µ$\varepsilon$, the displacement was returned slowly until the specimen had original length. This cycle was done seven times to obtain reliable results. The electrical and mechanical strains were simultaneously recorded during the test to investigate the piezoresistive behavior in the CFRP.

Figure 5 shows a schematic electrical circuit used to record the strains. The circuit used two Wheatstone bridges. The first bridge was connected to the strain gauge (TML FLA-5-11) to measure the mechanical strain in the CFRP. The second bridge was connected to two electrodes to measure the electrical strain alteration. In the second bridge, one dummy strain gauge was installed to assist the balancing of the bridge circuit resistances. This had to be done because the electric resistance of the CFRP is very small compared to the Wheatstone bridge resistances. The dummy strain gauge was pasted to a sufficiently rigid steel plate. Thus, no electrical resistance alteration generated by the dummy strain gauge. Both strain gauges attached in the CFRP and steel plate (as dummy strain gauge) have gauge factor ($G_{sg}$) of 2.09, electrical resistance ($R_{sg}$) of 120 ± 0.3 Ω, and a gauge length of 10 mm. By using the bridges, the recorded data was voltage signal alteration only. This method, which used a Wheatstone bridge, was more beneficial than the four-wire method because the voltage change was usually small compared to initial resistance at the CFRP. As a consequence, the resistant alteration data obtained in the method would be more accurate.

The voltage signals from each Wheatstone bridge are then filtered to eliminate noise signals and amplified by a strain amplifier TML DA-37A. From the first Wheatstone bridge circuit, the mechanical strain ($\varepsilon_m$) in CFRP can be evaluated by calculating Equation (10) as follows,

$$\varepsilon_m = \frac{V_m \cdot K_m}{G_{sg}}, \tag{10}$$

where $V_m$ is DC voltage signal recorded from the first Wheatstone bridge circuit and $K_m$ is an amplified voltage factor that can be adjusted in the first strain amplifier. From the second Wheatstone bridge circuit, an electrical strain ($\varepsilon_e$) can be then calculated as follows,

$$\varepsilon_e = V_e K_e, \tag{11}$$

where $V_e$ is DC voltage signal recorded from the second Wheatstone bridge circuit and $K_e$ is an amplified voltage factor that can be adjusted in the second strain amplifier. To obtain the optimal amplification of the voltage signal, the value of $K_m$ is set to 1000 µε/V for the first Wheatstone bridge circuit, whereas for the second Wheatstone bridge circuit, the value of $K_e$ is set to 50 µε/V. Furthermore, the voltage signals were also converted into a digital signal by the ADC prior to one displayed in the data recorder. Note that for each cyclic tensile test, the voltage signals were recalibrated. Thus, the signals always start from zero before the woven CFRP specimen was imposed by the tensile loading.

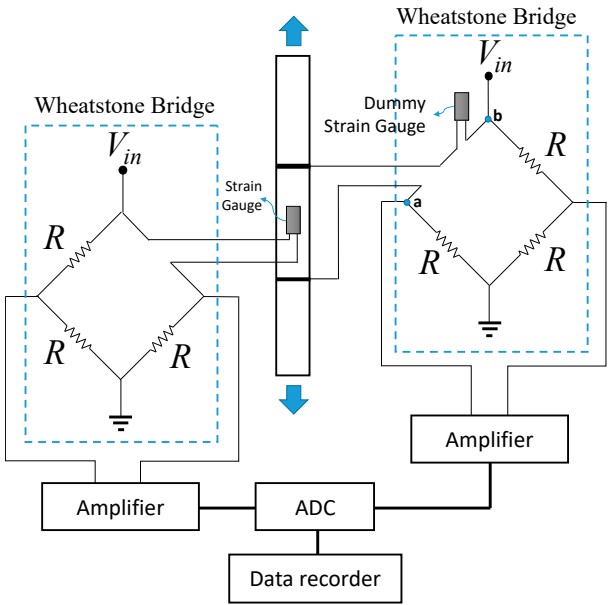

**Figure 5.** Schematic electrical circuit installed during tensile tests.

Considering points *a* and *b* in Figure 5, the resistance between those points ($R_{ab}$) can be calculated as follows,

$$R_{ab} = R_{CFRP} + 2R_c + R_{sg}, \tag{12}$$

The $\varepsilon_e$ in Equation (7) can be derived by considering the alteration of electrical resistance recorded in the data recorder, which is sourced by the mechanical strain of the CFRP only. This consideration is reasonable since the $R_c$ and $R_{sg}$ do not contribute to electrical resistance changes. Combining Equations (1), (7) and (12), the $\varepsilon_e$ can be then calculated as follows,

$$\varepsilon_e = \frac{V_e K_e R_{ab}}{R_{CFRP}}, \tag{13}$$

## 4. Results and Discussion

Figure 6a,b show $V_{DC}$ versus $I_{DC}$ curves in wrap and thickness directions, respectively. The linear relationship between $I_{DC}$ and $V_{DC}$ for both directions is revealed from 17 different $I_{DC}$ values. The gradients of the regression curve of the results represent $R_{Total}$ values of the woven CFRP, which are then plotted against $L_c$, the distance between the two electrode points, as depicted in Figure 7. Note that the $R_{Total}$ measured here uses the four-wire method, and the woven CFRP specimen was not imposed by tensile loading.

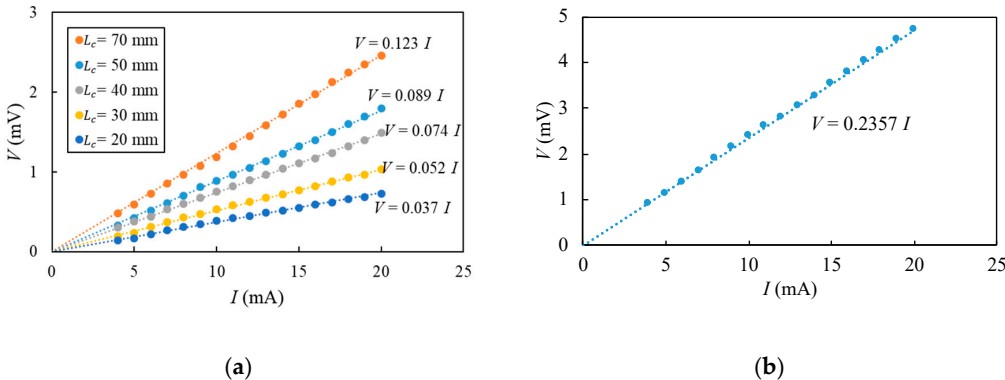

**Figure 6.** Voltage to current curves in woven CFRP for (**a**) wrap direction and (**b**) thickness direction.

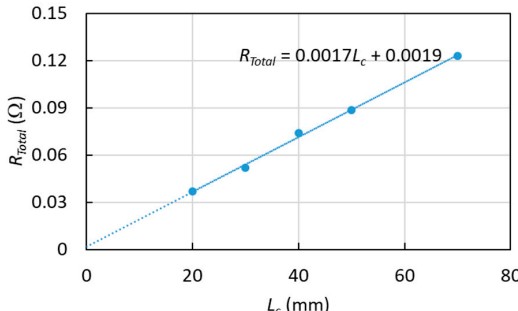

**Figure 7.** Relationship between total electrical resistance and two electrodes distance for 6 laminas of plain-woven composite.

For the wrap direction, the $R_{Total}$ values can be plotted against $L_c$ as shown in Figure 8. The $R_{Total}$ increased proportional with $L_c$ which indicates the geometry and electrical properties of the woven CFRP were relatively uniform. Linear regression can then be drawn from the five plotted points. The $L_c$ equals to zero gives $R_{Total}$ of 0.0019 $\Omega$ which equals to $2R_c$. Thus, the $R_c$ of 0.00095 $\Omega$ was obtained. This $R_c$ value was relatively small which indicates the attachment method of the electrode to the woven CFRP surface was effective. For thickness direction, the $R_{Total}$ of 0.2357 $\Omega$ was measured. Considering that the woven CFRP has sid laminae and the appearance of $R_c$, the $R_t$ of 0.039 $\Omega$ was obtained by calculating Equation (3). Furthermore, by calculating Equation (2), $R_w$ with a distance of 70 mm, can be obtained, i.e., 0.307 $\Omega$.

Electrical resistivity ($\rho$) can then be calculated by using Equations (4) and (5). The values of $\rho_{w(6)}$ and $\rho_{t(6)}$ are shown in Figure 9. The experiment for the wrap direction was conducted for five different electrode distances. The span of $\rho_{w(6)}$ values was relatively small with an average value of 10.1 $\mu\Omega$m. In thickness direction, $\rho_{t(6)}$ of 43.8 $\mu\Omega$m was obtained. The results show $\rho_{t(6)}$ was four times higher than $\rho_{w(6)}$. This occurs because in the thickness direction, there were five interfaces that only contain epoxy resin. However, in wrap direction, the carbon fiber acts as an electric conductor with relatively small resistance.

After the specimens have been tested and the results calculated using collected data, it was found that the contact resistance of CFRP will decrease in accordance with its polishing treatment, as we can see from Figure 9. The higher the grit number of the polishing paper used, the surface roughness decreases, and the contact resistance also decreases. High grit number in the polishing paper used makes the surface of the specimen smoother and increases the contact area of electrode and carbon fibers in the specimens, where high contact area decreases contact resistance in CFRP. Lower contact resistance is better for resistance measurement of the CFRP because if the contact resistance is high, it can hinder the total resistance measurement of the CFRP that are very small. For the tensile test, based on the data acquired as depicted in Figure 9b, it may be inferred that the polishing process did

not significantly affect CFRP strength. The above remark is supported by the fact that the unpolished specimen is not necessarily stronger than the polished specimens.

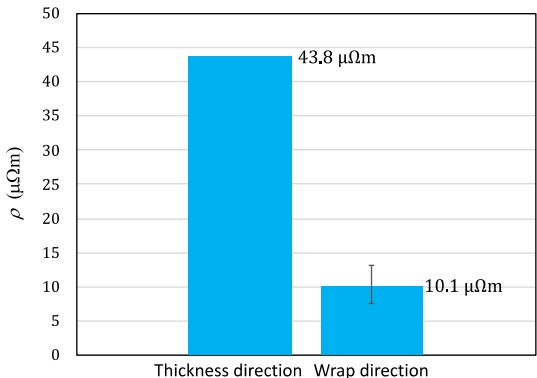

**Figure 8.** Electrical resistivity of plain-woven CFRP in thickness and wrap directions.

Tensile tests are conducted for specimens installed electrical circuit shown in Figure 5. In the tensile tests, the specimen is loaded and unloaded for seven times. The example voltage signals ($V_m$ and $V_e$) obtained for one cycle loading and unloading can be seen in Figure 10. The vertical axis in Figure 10 shows the $V_m$ and $V_e$ obtained from the data recorder. It can be seen that the $V_e$ increases as the response of increased $V_m$ and vice versa. However, the $V_e$ the response is not perfectly stable. This can occur due to the movement of fiber arrangements inside the plain-woven CFRP, which might not be a linear response as discussed in sub-Section 2.1. To investigate this nonlinearity further, $V_m$ is then converted to $\varepsilon_m$ by using Equation (10), whereas the $V_e$ is converted to $\varepsilon_e$ by using Equation (11).

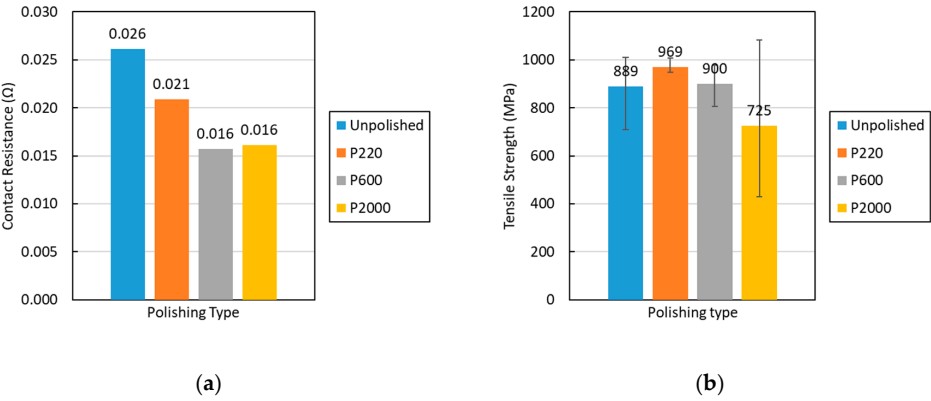

(**a**)          (**b**)

**Figure 9.** (**a**) contact resistance values and (**b**) tensile strength of plain-woven CFRP for different polishing treatment.

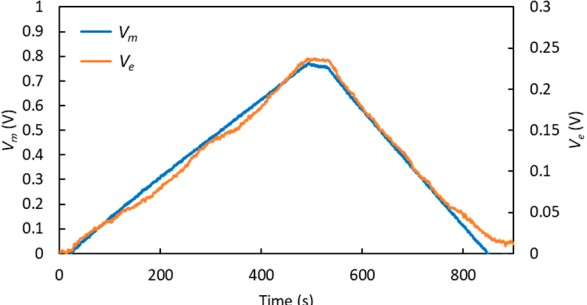

**Figure 10.** Example of voltage signals obtained from the strain gauge and woven CFRP under loading and unloading conditions.

Figure 11a shows one example of data processing of voltage signals ($V_m$ and $V_e$) to $\varepsilon_m$ and $\varepsilon_e$. It may be observed that the residual $\varepsilon_e$ appears when $\varepsilon_m$ equals 0 in the unloading process. The results of all seven data processing of voltage signals are then plotted as shown in Figure 11b,c. From the figure, it can be observed that $\varepsilon_e$ is proportional to $\varepsilon_m$. This proves that the positive piezoresistivity in the woven CFRP can be observed with Wheatstone bridge circuits, similar to unidirectional CFRP as reported by Todoroki et al. [8]. However, for the plain-woven composite subjected by loading, the variance of $\varepsilon_e$ becomes higher when the higher loading is applied. In particular, the variance is clearly seen when the $\varepsilon_m$ reaches 0.0002. This may occur because when the loading is applied, the fibers having plain-woven pattern have more different possible arrangements which directly determine the $R_{CFRP}$. Upon unloading, plain-woven composite only displays a small reduction of the variance. There is hysteresis appearing in the measurement. This result indicates the fibers do not return back to the initial position after unloading. Thus, residual $\varepsilon_e$ of 0.0016 was observed when $\varepsilon_m$ returned back to 0.

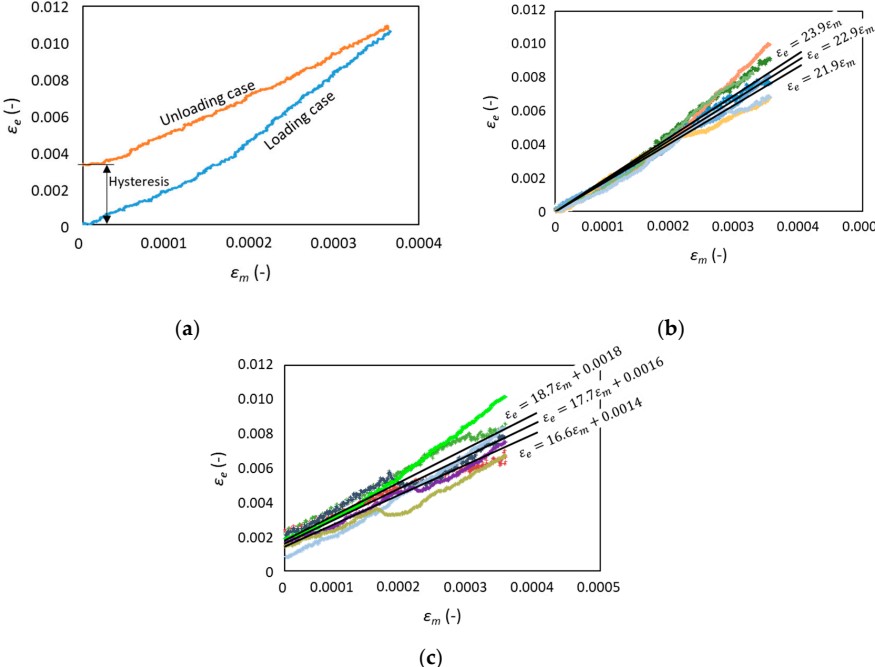

**Figure 11.** Electrical and mechanical strains relationship for plain-woven composite: (**a**) example of one loading and unloading case, (**b**) seven test results of specimens subjected to loading and (**c**) unloading.

For SHM development, $G_{CFRP}$ is a parameter that needs to be investigated. The variance of the $G_{CFRP}$ must be considered in the development. In this work, the linear least square method is used to analyze the data obtained because this method allows researchers to study how the variables are connected and related by placing the line's equation as close as possible to all data. In Figure 11, $\varepsilon_m$ is the independent random variable or predictor and $\varepsilon_e$ is the responsible variable. The linear least square equation shows how both variables are connected in a mathematical way. Here, confidence interval theory, usually expressed in percentage, is also used to ensure the processed data is statistically valid. For example, the commonly used interval, 95% confidence interval is a range of values that the researcher can be 95% certain it contains the true mean of the population or when the experiment is repeated over and over again, 95% of the results will match with the results from the population. Thus, by using this method, it will ensure the obtained and processed data is statistically valid.

It is important to note that the calculation is divided into two cases, loading case and unloading case. This has to be done because there are different characteristics appearing which depends on the loading condition. Table 1 shows a summary of the gauge factor calculation. The gauge factor during loading conditions is higher than the unloading condition.

**Table 1.** Electrical properties of plain-woven CFRP.

| Parameter | Loading Condition | Unloading Condition |
|---|---|---|
| $G_{CFRP}$ range | 21.9–23.9 | 16.6–18.7 |
| Median $G_{CFRP}$ | 22.9 | 17.7 |
| Averaged residual electric strain | - | 0.0016 |

The occurrence of electric strain variance must be considered in SHM development for plain-woven CFRP composite. This can be conducted by recording resistance data based on time series. The recorded data must include loading and unloading status.

## 5. Conclusions

Positive piezoresistivity has been observed in plain-woven CFRP materials using a Wheatstone bridge circuit which shows that electrical strain changes are proportional to mechanical strain changes. The positive piezoresistivity is reversible based on the results of observations. When the material experiences unloading conditions or the decrease in mechanical strain the value of the electrical strain also decreases. However, the electric strain value does not return to zero or hysteresis. From the experimental results, the CFRP woven fiber specimen has an electric resistivity of 10.1 μΩm in the wrap direction and 43.8 μΩm in the direction of thickness. The value of the gauge factor of plain-woven CFRP material is 22.9 for loading conditions and 17.7 for unloading conditions.

In developing structural health monitoring for plain-woven composite, the gauge factor values obtained in this work are useful to accurately predict mechanical strain condition in the composite. In this case, the recorded electrical strain must be observed together with the history of loading conditions, including typical loading or unloading, which determines the gauge factor value. By considering the electrical strain and the typical loading simultaneously, the mechanical strain condition can be accurately predicted. Furthermore, the accurate prediction of mechanical strain can assist us in detecting damages in the composite, which usually redistributes the strain condition. In addition, the relation between mechanical strain and the existence of damage in the composite may be explored in future work.

**Author Contributions:** Conceptualization, I.P.N. and B.A.B.; methodology, I.P.N.; validation, I.P.N., B.A.B., and M.A.; formal analysis, I.P.N., B.A.B., and M.A.; investigation, A.A.A., S.T.U., P.N.H.; writing—original draft preparation, I.P.N., B.A.B., A.A.A., and S.T.U.; writing—review and editing, P.N., M.A.; visualization, B.A.B. and A.A.A.; supervision, I.P.N., B.A.B., and M.A.; project administration, B.A.B.; funding acquisition, I.P.N. and B.A.B. All authors have read and agreed to the published version of the manuscript.

**Funding:** This research is funded by ITB under Research, Innovation, and Community Services Program (P3MI) of the year 2019, USAID through Sustainable Higher Education Research Alliances (SHERA) program under grant number IIE00000078-ITB-1, and an undergraduate research grant from Faculty of Mechanical and Aerospace Engineering.

**Conflicts of Interest:** The authors declare no conflict of interest.

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
