# Peer review of "Nonlinear Piezoresistive Behavior of Plain-Woven Carbon Fiber Reinforced Polymer Composite Subjected to Tensile Loading"

_applsci, doi:10.3390/app10041366_

Round 1

Reviewer 1 Report

Dear authors! Please, consider these suggestions and questions.

73 - 75. A novel method to investigate the relationship between electrical and mechanical strains by using a Wheatstone bridge circuits is proposed. Wheatstone bridge technique cannot be named as "novel".

92 - 93. But there are no convincing results of the study of CFRP cyclic loading

are attached? Or a single electrode?

Formulas (2) - (5), Figure 2 and rows 137 - 138. Was the assumption of non-linearity and hysteresis of the material used, or its behavior was assumed linear?

142-143. Please, inform еру reader about structural anizotropy of specimen's material. Isotropic? Transversely isotropic? Orthotropic?

144-146. Please explain whether it is necessary to polish the entire surface of the composite part under test at the operating conditions for its structural health monitoring? Probably the proposed method is more suitable for the quality control during composite parts manufacturing, i.e. for NDE?

Figure 4, b is not clear.

185-186. The authors don't inform us about strain gauge calibration before the specimens testing. How the strain was measured originally? By the cross head displacement or by using extensometer?

Figure 5. It's unclear, why only one strain gauge was used to measure the tensile strain. The use of two identical strain gauges, which are placed to the opposite branches, is more reliable for the measure the gauge's resistance dependence on the strain.

255-256. The authors conclude: "But, from these data, we can conclude that the polishing process didn’t significantly affect CFRP strength". Whereas the previous sentence is: "This means that for this test, more sample will be needed to get a quantitative relation between polishing treatment to the strength of CFRP."

Moreover, regarding Figure 9,b we cannot understand, which statement is true?

Caption of Figure 11 should be corrected.

281. The sentence " There is hysteresis appearing in the measurement" should be illustrated in some additional plots with greater visibility because Figure 11 demonstrates no hysteretic behavior of the strain, but a low stability of the measurements results.

Conclusion. Unfortunately, the authors reported the values of CFRP resistance only in Conclusions. No information about resistivity (Ohm*m) in both directions. No information about sensitivity of these values to the strain (Ohm*m/ε) and about their confidence intervals.

Two very important questions arise after reading this paper.

First question. What inconsistencies or defects of the composite materials the authors want to detect by using CFRP resistance measuring?

Second question. The authors throughout the article use the term "piezoelectric resistance", but no data are given on the piezoelectric effect. The dependence of the electrical resistance of a material on tensile strain is investigated.

Your paper requires revision.

Author Response

Reviewer 1

Dear authors! Please, consider these suggestions and questions.

Dear Reviewer, thank you for reviewing our article. Your comment and suggestion are valuable to improve our article quality for wide readers.

73 - 75. A novel method to investigate the relationship between electrical and mechanical strains by using a Wheatstone bridge circuits is proposed. Wheatstone bridge technique cannot be named as "novel".

Thank you for your suggestion. We integrate Wheatstone bridge circuit and electrodes for measuring electrical resistance of the composite. This integration is different with common method to measure electrical resistance of the CFRP i.e. by using two-wire method or four-wire method. However, as your concern, this integration cannot be called as ‘novel’ method. Thus, we delete word ‘novel’ and revise the sentence as shown in line 74,

“A method to investigate the relationship between electrical and mechanical strains is proposed by integrating Wheatstone bridge circuits, strain gauge, and electrodes. This method is different as compared to the common two or four wire methods.”

92 - 93. But there are no convincing results of the study of CFRP cyclic loading

are attached? Or a single electrode?

Thank you for your comment. As for the cyclic loading, we perform loading and unloading seven times for the same specimen. The explanation in line 92 - 93 is based on the hypothesis that carbon fiber has high electrical conductivity and the fiber arrangement of the plain-woven composite will not come back to initial position during unloading process. Even though, fiber arrangement determines the overall electrical resistance of the plain-woven CFRP. This hypothesis is proven in experimental results as shown in Figure 11a and 11b as follows,

(a)

(b)

Figure 11. electrical and mechanical strains relationship for plain-woven composite: (a) subjected to load and (b) released by load

We have added an explanation regarding the experimental results which proves the hypothesis as shown in line 278,

“The results of seven data processing of voltage signals ( and ) to  and  are then plotted as shown in Figure 11. From the figure, it can be observed that  is proportional with . This proves that the positive piezoresistivity in the woven CFRP can be observed with Wheatstone bridge circuits, similar with unidirectional CFRP as reported by Todoroki et al. [8]. However, for the plain-woven composite subjected by loading, the variance of  becomes higher when the higher loading is applied. In particular, the variance is clearly seen when the  reaches 0.0002. This may occur because when the loading is applied, the fibers having plain-woven pattern have more different possible arrangements which directly determine the . Upon unloading, plain-woven composite only displays a small reduction of the variance. There is hysteresis appearing in the measurement. This result indicates the fibers do not return back to the initial position after unloading. Thus, residual  of 0.0016 is observed when  returns back to 0”

As for single electrode comment, in the specimen, we attach only a pair of cable to the polished surface of the CFRP. Therefore, the specimen is indeed a single electrode.

Formulas (2) - (5), Figure 2 and rows 137 - 138. Was the assumption of non-linearity and hysteresis of the material used, or its behavior was assumed linear?

Thank you for your question. These equations are built based on model in Figure 2 in which the resistance in longitudinal and transversal directions of all layers are assumed to be uniform. Therefore, those equations do not reflect non-linearity and hysteresis of the plain-woven composite. The non-linearity and hysteresis are reflected by different gauge factor value, which is investigated when the composite is subjected to loading and unloading. The results show that the gauge factor, which is the most important electrical property of the material for SHM application, has different value depending on the loading case.

Regarding your concern on the model in rows 137 – 138, We have revised the explanation as shown in line 136.

“Positive value of  indicates positive piezoresistive behavior and vice versa. In case that nonlinear piezoresistive behavior appears, the  under loading and unloading conditions would have different values. This difference must be investigated experimentally for developing SHM. Selection of  value applied to the electrical resistance model of the plain-woven composite must consider the nonlinearity and the existence of residual resistance.”

142-143. Please, inform еру reader about structural anizotropy of specimen's material. Isotropic? Transversely isotropic? Orthotropic?

Thank you for your suggestion. The plain-woven material has orthotropic material properties. We have revised our manuscript to add this information as shown in line 146,

“The CFRP is prepared by cutting the plate to follow the geometry of the tensile test specimen as depicted in Figure 3. Since the CFRP has orthotropic properties, the cutting line must be parallel to the fiber wrap direction.”

144-146. Please explain whether it is necessary to polish the entire surface of the composite part under test at the operating conditions for its structural health monitoring? Probably the proposed method is more suitable for the quality control during composite parts manufacturing, i.e. for NDE?

Thank you for your suggestion. The polish treatment is only necessary in area that will be installed the electrodes. We have added this information as shown in line 149,

“Prior to the installation, the surface in which we will connect the electrodes is first polished to minimize contact resistance between the electrodes and the specimen.”

The proposed method may detect strain alteration in the structure which occurs during manufacturing process of the composite or when damage appears. However, in this manuscript, we focus on investigating the relationship between electrical and mechanical strains in the plain-woven composite, which is the key to apply the method for SHM development.

Figure 4, b is not clear.

Thank you for your comment. We have revised Figure 4 as follows,

(a)

(b)

Figure 4. Electrical circuit for initial resistance evaluation in (a) wrap and (b) thickness directions

185-186. The authors don't inform us about strain gauge calibration before the specimens testing. How the strain was measured originally? By the cross head displacement or by using extensometer?

Thank you for your comment. Strain is measured by strain gauge and a standard set of strain measuring equipment. In this case, the strain gauge is connected to strain amplifier through the bridge completion set. The strain amplifier is factory calibrated and also the strain gauge. In the setting, we adjust such that one volt equals to 1000 microstrains. Since the strain gauge uses standard measuring equipment, the measured strain is considered accurate.

We have revised our manuscript to add explanation regarding the standard strain gauge and strain measuring equipment as shown in line 190 and 202,

“The first bridge is connected to the strain gauge (TML FLA-5-11) to measure mechanical strain in the CFRP.”

“The voltage signals from each Wheatstone bridge are then filtered to eliminate noise signals and amplified by a strain amplifier TML DA-37A.”

We have also added information regarding the instrumentation setting as shown in line 209,

“To obtain the optimal amplification of voltage signal, the value of  is set to 1000 μe/V for the first Wheatstone bridge circuit, whereas for the second Wheatstone bridge circuit, the value of  is set to 50 μe/V.”

Figure 5. It's unclear, why only one strain gauge was used to measure the tensile strain. The use of two identical strain gauges, which are placed to the opposite branches, is more reliable for the measure the gauge's resistance dependence on the strain.

Thank you for your comment. As your concern, the use of two strain gauges on both sides of the CFRP plate will increase the sensitivity of strain measurement. However, since we have to attach the electrodes on the other side of the plate, then we can only use one strain gage to measure the tensile strain. As the result, the measured strains obtained by strain gauge and electrodes are comparable (‘apple to apple’).

255-256. The authors conclude: "But, from these data, we can conclude that the polishing process didn’t significantly affect CFRP strength". Whereas the previous sentence is: "This means that for this test, more sample will be needed to get a quantitative relation between polishing treatment to the strength of CFRP."

Thank you for your correction. We have conducted experiment to reveal the relationship between polish treatment and CFRP strength. the conclusion is, polish treatment does not influence the CFRP strength. we have revised our manuscript as shown in line 256,

“For the tensile test, based on the data acquired as depicted in Figure 9b, it may be inferred that the polishing process did not significantly affect CFRP strength. The above remark is supported by the fact that the unpolished specimen is not necessarily stronger than the polished specimens.”

Moreover, regarding Figure 9b we cannot understand, which statement is true?

Caption of Figure 11 should be corrected.

Thank you for your correction. We have subsequently revised the sentence as shown in line 256.

We have also revised caption of Figure 11 as follows,

“Figure 11. Electrical and mechanical strains relationship for plain-woven composite: (a) subjected to loading and (b) unloading”

The sentence " There is hysteresis appearing in the measurement" should be illustrated in some additional plots with greater visibility because Figure 11 demonstrates no hysteretic behavior of the strain, but a low stability of the measurements results.

Thank you for your suggestion. To find the hysteresis, we have to compare loading and unloading condition. Ideally, the two graphs for loading and unloading would be placed in one frame. However, considering the variation of the values, it is not possible for us to merge the two graphs. The hysteresis is indicated by the existence of the residual electrical strain while the mechanical strain equals to 0.

We have revised our manuscript to explain the above as shown in line 285,

“There is hysteresis appearing in the measurement. This result indicates the fibers do not return back to the initial position after unloading. Thus, residual  of 0.0016 is observed when  returns back to 0.”

Conclusion. Unfortunately, the authors reported the values of CFRP resistance only in Conclusions. No information about resistivity (Ohm*m) in both directions. No information about sensitivity of these values to the strain (Ohm*m/ε) and about their confidence intervals.

Thank you for your comment. The information of resistivity can be found in Figure 8. We have also added information regarding the value as shown in line 239,

“The experiment for wrap direction is conducted for five different electrode distances. The variance of  is in small range with average value of 1.01 x 105 m. In thickness direction, the  of 4.38 x 105 Wm is obtained. The results show  is 4 times higher than . This occurs because in thickness direction, there are five interfaces which only contain epoxy resin. However, in wrap direction, the carbon fiber acts as electric conductor with relatively small resistance.”

As your concern, the sensitivity measurements were performed in limited number of repetitions and thus do not allow us to present in terms of confidence intervals.

Two very important questions arise after reading this paper.

First question. What inconsistencies or defects of the composite materials the authors want to detect by using CFRP resistance measuring?

Thank you for your question. This paper focuses on the finding of the relationship between electrical strain and mechanical strain only. In addition, in this paper, we focus on revealing the nonlinear behavior of the piezoresistivity during loading and unloading cases indicated by gauge factor difference. This parameter is important to accurately determine mechanical strain condition which directly reflect the damage condition in the composite. At the moment, we are investigating the piezoresistivity of defective composite. So far we have not gotten a comprehensive result.

Second question. The authors throughout the article use the term "piezoelectric resistance", but no data are given on the piezoelectric effect. The dependence of the electrical resistance of a material on tensile strain is investigated.

Thank you for your question. We did not use the term ‘piezoelectric resistance’ but piezoresistivity. This term of piezoresistivity is defined as the change of electrical resistance of two points of the composite due to mechanical strain. The resistance comes from the fiber component of the specimen, i.e., the carbon fiber. Our understanding is that SHM community simply adopts the term piezoresistivity to CFRP composite due to its similarity, .i.e, introduction of strain changes results in resistivity changes. In terms of number, the resistance change in the composite due to loading is relatively small, unlike piezoresistivity in piezo material.

Your paper requires revision.

Thank you very much. We have conducted major revision throughout out paper. We strongly expect that the revision that we have done can fulfill the standard for publication in this journal.

Reviewer 2 Report

This manuscript is in good shape to be published. Only a few points need to be addressed.

please provide the details of the plain-woven CFRP in this study, what is the manufacturer?  please provide the details of the epoxy resin, brand and etc.  check the axis label in Figure 8.

Author Response

Reviewer 2

This manuscript is in good shape to be published. Only a few points need to be addressed.

Dear Reviewer, thank you for reviewing our article. Your comment and suggestion are valuable to improve our article quality for wide readers.

please provide the details of the plain-woven CFRP in this study, what is the manufacturer?  please provide the details of the epoxy resin, brand and etc.

Thank you for your suggestion. We purchased the CFRP plate through an online merchant and it is made in China. However, no detailed specifications are provided.

Check the axis label in Figure 8.

We have also revised the axis in Figure 8 as follows,

Figure 8. Electrical resistivity of plain-woven CFRP for (a) thickness and (b) wrap directions

Reviewer 3 Report

In the manuscript entitled "Nonlinear piezoresistive behavior on plain-woven carbon fiber reinforced polymer composite subjected to tensile loading" by Nurprasetio et al. investigates the relationship between electrical and mechanical strains by using a Wheatstone bridge circuits.

Overall, the manuscript provide quite interesting information. Te experiments are well-designed and the results are clearly presented. Figures are very helpful. 

However the authors should use more relevant literature and empasize the novelty of their study.

Author Response

Reviewer 3

In the manuscript entitled "Nonlinear piezoresistive behavior on plain-woven carbon fiber reinforced polymer composite subjected to tensile loading" by Nurprasetio et al. investigates the relationship between electrical and mechanical strains by using a Wheatstone bridge circuits.

Dear Reviewer, thank you for reviewing our article. Your comment and suggestion are valuable to improve our article quality for wide readers.

Overall, the manuscript provide quite interesting information. Te experiments are well-designed and the results are clearly presented. Figures are very helpful.

However the authors should use more relevant literature and empasize the novelty of their study.

Thank you for your suggestion. We have added more references to emphasize the novelty of our research as shown in references 18 and 19.

Ogihara, S.; Reifsnider, K.L. Characterization of nonlinear behavior in woven composite laminates. Applied Composite Materials 2002, 9, 249-263. Ishikawa, T.; Chou, T.W. Nonlinear behavior of woven fabric composites. Journal of Composite Materials 1983, 17, 399-413.

Round 2

Reviewer 1 Report

Dear authors! Your article is sufficiently improved, but there are some questions and suggestions.

In order to improve a visibility of Fig. 11. I recommend to split some loading and unloading curves from the Fig. 11, a,b, then connect 2-3 curves, which will demonstrate the loading-unloading. Too many curves are present together. It's impossible to separate at least one hysteresis loop V(ε) when ε changes from zero to the positive value, then to the negative value and so one. It can visually confirm the hysteretic behavior of the strain-resistance dependence. I should repeat my question, which was given in my 1st review: " How the strain gauges were calibrated? ". But I did not get a clear answer. So, I will try to explain my question with more details. When strain gauge is deformed (is stretched out) its resistance is changed. If this gauge is connected to the balanced bridge, the unbalancing voltage will arise on the measuring diagonal. This voltage is amplified by amplifier, then transformed by the AD converter and stored as some value in a file. If the measuring setup has a very good precision and sensitivity, it's nice. But the measured value is the ELECTRIC value - not the mechanical strain. In order to know the strain experienced by the strain gauge we need to have some mean to know the MECHANICAL value of strain. There are two ways to determine the mechanical strain: measurement of the cross-head displacement, which is allowed by all modern mechanical testing machines. But this way is not good because the testing machine is deformed itself during loading, and its deformation distorts the measuring results. The second, the best way, is when specimen deformation (change of the distance between two base points) is measured by the extensometer, which sensitivity is provided by its manufacturer. What way was used in your study? Please, confirm your definition of the resistivity. Its definition, which is accepted everywhere is the following. Let the electric wire of length L and its cross-section area is S has the resistance R. This resistance is equal R=ρ*L/S, where ρ is the wire's material resistivity. My question arises because you present the enough big values for CFRP resistivity.

4. In the paper's conclusion you present some numerical data obtained as the results of your study. Please, justify, why these results can be effectively used at the design of SHM (sensitivity, ability to detect some material inconsistencies etc.). Here you can used even some guess-work. Some information about your future work in this direction also is desirable.

Author Response

Dear authors! Your article is sufficiently improved, but there are some questions and suggestions.

Dear Reviewer, thank you for the excellent review on our paper and the encouraging words. Your comments and suggestions are invaluable in improving our article.

In order to improve a visibility of Fig. 11. I recommend to split some loading and unloading curves from the Fig. 11, a,b, then connect 2-3 curves, which will demonstrate the loading-unloading. Too many curves are present together. It's impossible to separate at least one hysteresis loop V(ε) when ε changes from zero to the positive value, then to the negative value and so one. It can visually confirm the hysteretic behavior of the strain-resistance dependence.

Thank you for your suggestion. We have added Figure 11.a as one example of ‘line curves’ for the loading and unloading cases. We then combine seven data sets from loading cases as depicted in Figure 11.b and seven data from unloading cases as depicted in Figure 11.c.

(a)

(b)

(c)

Figure 11. Electrical and mechanical strains relationship for plain-woven composite: (a) example of one loading and unloading case, (b) seven test results of specimens subjected to loading and (c) unloading

We have also performed important correction for the estimation of the gauge factor in the loading case. The regression line must intersect point (0, 0). We have also revised the gauge factor value of the loading case as shown in line 284,

“Figure 11a shows one example of data processing of voltage signals ( and ) to  and . It may be observed that the residual  appears when  equals 0 in unloading process. The results of all seven data processing of voltage signals are then plotted as shown in Figure 11b and 11c.”

And line 324,

“From the experimental results, the CFRP woven fiber specimen has an electric resistivity of 10.1 mWm in the wrap direction and 43.8 mWm in the direction of thickness. The value of the gauge factor of plain-woven CFRP material is 22.9 for loading condition and 17.7 for unloading condition.”

We have also revised Table 1 as follows,

Table 1. Electrical properties of plain-woven CFRP.

parameter

Loading condition

Unloading condition

 range

21.9 – 23.9

16.6 – 18.7

Median

22.9

17.7

Averaged residual electric strain

-

0.0016

I should repeat my question, which was given in my 1st review: " How the strain gauges were calibrated? ". But I did not get a clear answer. So, I will try to explain my question with more details. When strain gauge is deformed (is stretched out) its resistance is changed. If this gauge is connected to the balanced bridge, the unbalancing voltage will arise on the measuring diagonal. This voltage is amplified by amplifier, then transformed by the AD converter and stored as some value in a file. If the measuring setup has a very good precision and sensitivity, it's nice. But the measured value is the ELECTRIC value - not the mechanical strain. In order to know the strain experienced by the strain gauge we need to have some mean to know the MECHANICAL value of strain. There are two ways to determine the mechanical strain: measurement of the cross-head displacement, which is allowed by all modern mechanical testing machines. But this way is not good because the testing machine is deformed itself during loading, and its deformation distorts the measuring results. The second, the best way, is when specimen deformation (change of the distance between two base points) is measured by the extensometer, which sensitivity is provided by its manufacturer. What way was used in your study? Please, confirm your definition of the resistivity. Its definition, which is accepted everywhere is the following. Let the electric wire of length L and its cross-section area is S has the resistance R. This resistance is equal R=ρ*L/S, where ρ is the wire's material resistivity. My question arises because you present the enough big values for CFRP resistivity.

Thank you for your extremely valuable correction. We found an (unacceptable) error when displaying values of resistivity of the composite material. The values of resistivity should be 1.01 x 10-5 Wm and 4.38 x 10-5 Wm or by the use of prefix, they are 10.1 mWm and 43.8 mWm. For some reason, the power becomes positive in the composition process. Relying on your comment, we returned to the research reports and found that the unit is in 10-5 instead of 105. We have revised our manuscript accordingly as shown in line 245,

“The span of values is relatively small with average value of 10.1 mWm. In thickness direction,  of 43.8 mWm is obtained.”

We have also revised Figure 8 to correct the unit to be micro-ohm meter as follows,

Figure 8. Electrical resistivity of plain-woven CFRP in (a) thickness and (b) wrap directions

As your concern about calibration, we did not perform any calibration because we use factory (test result) gauge factor () of 2.09. The mechanical strain is simply calculated by using equation 10 as follows,

Based on our experience of using strain gauge, for standard application of the strain gauge, we simply use the factory calibrated Gauge Factor (comes with the packaging). We did use extensometer in the past, but it was performed in Young’s modulus estimation.

In the paper's conclusion you present some numerical data obtained as the results of your study. Please, justify, why these results can be effectively used at the design of SHM (sensitivity, ability to detect some material inconsistencies etc.). Here you can used even some guess-work. Some information about your future work in this direction also is desirable.

Thank you for your suggestion. We have revised our manuscript as shown in Line 324,

“From the experimental results, the CFRP woven fiber specimen has an electric resistivity of 10.1 m in the wrap direction and 43.8 m in the direction of thickness. The value of the gauge factor of plain-woven CFRP material is 22.9 for loading condition and 17.7 for unloading condition.

In developing structural health monitoring for plain-woven composite, the gauge factor values obtained in this work are useful to accurately predict mechanical strain condition in the composite. In this case, the recorded electrical strain must be observed together with the history of loading conditions, including typical loading or unloading, which determines the gauge factor value. By considering the electrical strain and the typical loading simultaneously, the mechanical strain condition can be accurately predicted. Furthermore, the accurate prediction of mechanical strain can assist us to detect damages in the composite, which usually redistributes strain condition. In addition, the relation between mechanical strain and the existence of damage in the composite may be explored as a future work.”

From the bottom of our heart, we highly appreciate your comments and discussions that leads us to the founding of the embarassing mistake. Thank you so much
